# This is the way: designing and compiling LEPISZCZE, a comprehensive NLP benchmark for Polish

**Łukasz Augustyniak**
WUST
(Wroclaw University of Science and Technology)

**Kamil Tagowski**
WUST

**Albert Sawczyn**
WUST

**Denis Janiak**
WUST

**Roman Bartusiak**
WUST

**Adrian Szymczak**
WUST

**Marcin Wątroba**
WUST

**Arkadiusz Janz**
WUST

**Piotr Szymański**
WUST

**Mikołaj Morzy**
Poznan University of Technology

**Tomasz Kajdanowicz**
WUST

**Maciej Piasecki**
WUST

## Abstract

The availability of compute and data to train larger and larger language models increases the demand for robust methods of benchmarking the true progress of LM training. Recent years witnessed significant progress in standardized benchmarking for English. Benchmarks such as GLUE, SuperGLUE, or KILT have become a *de facto* standard tools to compare large language models. Following the trend to replicate GLUE for other languages, the KLEJ benchmark[1] has been released for Polish. In this paper, we evaluate the progress in benchmarking for low-resourced languages. We note that only a handful of languages have such comprehensive benchmarks. We also note the gap in the number of tasks being evaluated by benchmarks for resource-rich English/Chinese and the rest of the world.

In this paper, we introduce LEPISZCZE [2], a new, comprehensive benchmark for Polish NLP with a large variety of tasks and high-quality operationalization of the benchmark. We design LEPISZCZE with flexibility in mind. Including new models, datasets, and tasks is as simple as possible while still offering data versioning and model tracking. In the first run of the benchmark, we test 13 experiments (task and dataset pairs) based on the five most recent LMs for Polish. We use five datasets from the Polish benchmark and add eight novel datasets. As the paper's main contribution, apart from LEPISZCZE , we provide insights and experiences learned while creating the benchmark for Polish as the blueprint to design similar benchmarks for other low-resourced languages.

---

[1]*klej* is the word for glue in Polish
[2]*lepiszcze* is the Polish word for glew, the Middle English predecessor of glue

36th Conference on Neural Information Processing Systems (NeurIPS 2022) Track on Datasets and Benchmarks.

# 1 Introduction

Lack of reproducibility is an infuriating problem in machine learning practice. The inability to reproduce evaluation results and conduct reliable model comparisons is usually related to poor code quality, unclear and cryptic selection of hyper-parameter values, the random introduction of multiple factors affecting classification performance, and lack of a well-defined evaluation protocol (Pineau et al., 2021). These problems can be circumvented by encouraging people to use standardized and peer-reviewed evaluation environments. The rapid development of diverse language technology has increased the need for reliable evaluation environments.

The reproducibility issues are intensifying even stronger as more novel language models emerge each year. We have seen a remarkable progress on many language understanding tasks, from language modeling (Brown et al., 2020; Rae et al., 2021; Hoffmann et al., 2022), Named Entity Recognition (Li et al., 2020; Ye et al., 2022), Q&A (Lan et al., 2020; Yang et al., 2019), or various text classification tasks (Peters et al., 2018; Bingyu and Arefyev, 2022) in recent years. Moreover, in the last decade, data-centric models have become the major direction in solving most problems in the NLP area. Researchers and industry experts focus more on curated datasets and their maintenance processes. Hence, benchmarking models based on many datasets and their various and constantly changing versions is a great challenge.

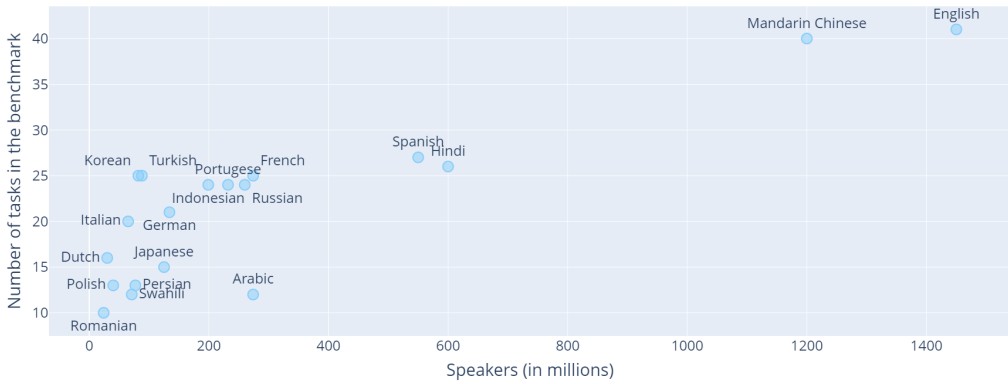

Figure 1: Tasks in large NLP benchmarks vs. the number of speakers, with Indian languages grouped for readability.

As shown in Figure 1, most NLP benchmarks are written for well-resourced languages such as English and Mandarin Chinese. This is understandable because many datasets exist in these languages, and many research teams are working in the context of these languages. Besides English and Mandarin Chinese, languages thoroughly covered with benchmarks include Indian, Spanish, French, and Portuguese. These are among the most commonly used languages in the world, and their position in the ranking is not surprising. However, we can also find Arabic and Japanese, widely spoken languages, but surprisingly few tasks are covered in benchmarks for these languages. Finally, we have languages with some benchmarks, such as Romanian, Persian, Dutch or Polish, but they only cover the most basic NLP tasks.

In this work, we focus on Polish and provide datasets and tools to facilitate research on Polish NLP tasks. We designed the benchmarking process so that it could be easily applied to other languages. Thus, preparing and adding benchmarks for other low-resourced languages should become much less laborious.

The Polish benchmarking tradition has a relatively short history. One of the few platforms for evaluating and comparing modern language models for Polish is the KLEJ benchmark (Rybak et al., 2020), a single-metric benchmark defined over a limited dataset. This simple practice for evaluating a model's performance no longer works. Current recommendations for the comparative evaluation of LMs advocate the inclusion of diversified tasks, challenges, and tests. Hence, we wanted to rethink and design a benchmark and environment to assess models so that they can still serve as valuable progress indicators.

Our main contributions are as follows:

- We propose LEPISZCZE , a new, extensive benchmark for Polish NLP with a large variety of tasks, expanding the previous Polish benchmark KLEJ with eight new datasets, published as a unified modern API.

- We design the benchmark and its maintenance using the best practices found in the literature, and we also investigate some of the most problematic aspects of creating benchmarks. We share the lessons learned while building the benchmark.

- We present the summary of training and evaluation of more than 6000 different models for LEPISZCZE , storing all information about code, dataset versions, parameters, metrics, predictions, or even information about their experimental environment.

## 2   Related Work

We used Google Scholar to review available NLP benchmarks for languages with at least 10 million speakers without going deeper into dialects (i.e., German includes all German dialects without dividing into Standard or Bavarian German). We searched for `<language name> NLP|NLG benchmark`. Furthermore, we dismissed benchmarks consisting of a single dataset. As a result, we found 35 benchmarks (see Table 1 in the Appendix), which included a total of 71 different tasks. Out of those, only 34 appeared in one language. These can be divided into two groups: specialized tasks which require a larger effort to build a good dataset (like diagnosis normalization, see (Wang et al., 2020)), or misdefined tasks such as Named Entity Recognition in the Polish KLEJ benchmark, which was not a span labeling task, but rather a text fragment classification task to detect if it contains an entity, without providing the span. We provide more detailed results of our survey in the supplementary materials.

When it comes to language coverage, only 31 languages have an existing NLP or NLG benchmark out of 91 available in the 2022 edition of Ethnologue (Fennig et al., 2022): Arabic, Assamese, Bengali, Chinese, Dutch, English, French, German, Gujarati, Hindi, Indonesian, Italian, Japanese, Kannada, Korean, Malayalam, Marathi, Odia, Persian, Polish, Portuguese, Punjabi, Romanian, Russian, Spanish, Swahili, Tamil, Telugu, Turkish, Urdu. The 74 tasks were not equally distributed per language, per Figure 1. Benchmarks for the two most commonly spoken languages: English and Chinese, would cover around 40 tasks. In contrast, the languages with the lowest number of tasks available in benchmarks and lower numbers of native speakers were Romanian and Polish (around 10 tasks). The results of our analysis are attached in Appendix.

The Polish language has a disproportionately small number of tasks in its main NLP benchmarks given nearly 40 million native speakers. KLEJ benchmark originally provided only 9 tasks, marking Polish the least task-covered European language concerning modern NLP and NLG benchmarked tasks. Once we take a deeper look at how tasks are formulated in KLEJ, we must acknowledge that the number of tasks formulated in a manner established in a given NLP sub-field is even smaller. There are two similar tasks of online reviews sentiment analysis, differing only in domains (PolEmo), and another sentiment analysis task (AR) framed as a regression task. Thus, the number of tasks can be reduced to 7 if we consider these datasets as one task. Moreover, some of the tasks are ill-defined. The NER task in KLEJ is not a sequence tagging but a document classification task. Summarization in KLEJ is evaluated based on classifying pairs of text and summary, the task is to predict whether the summary summarizes the text. Most benchmarks would define summarization as an NLG task, where the model is expected to generate the summary and would be evaluated with ROGUE or BLEU measures. A similar situation is happening in the Q&A task.

The KLEJ benchmark was created in 2020. Since then, no new datasets have been added, and the benchmark can be considered a little stale. Some of the tasks in KLEJ are not very difficult and diverse. Another potential problem with KLEJ is that it does not provide any environment for testing or submitting the model, as the submission requires only a prediction file. Finally, the heterogeneous, task-specific metrics for all the tasks in KLEJ could also be problematic when comparing the models as it may lead to erroneous conclusions. In our work, we aim to address these limitations. In particular, we plan to develop and maintain the long-term benchmark as part of the CLARIN-PL-Biz project.

# 3  LEPISZCZE

LEPISZCZE (IPA: **[lɛˈ pʲjʃʧɛ]** ) is an open-source benchmark and a continuous-submission leaderboard, concentrating public Polish datasets (existing and new) in specific tasks. Integrating datasets and tasks with model performance and efficiency allows academia and industry to gauge performance on tasks of interest quickly. Finally, it intends to foster constructive competition and innovation by bringing together and promoting previously disparate resources.

Our benchmark is structured into *datasets*, *tasks*, and *models*. We design the benchmark to be easily extendable and flexible so that leaderboards for various subsets of datasets, tasks, and models can be added in the future.

## 3.1  Benchmark designing and construction process

In this section, we introduce the benchmark construction process and lessons learned during this procedure, hoping that they could serve as a guide for other researchers that will face the task of creating a benchmark. The general design concept we follow is to make the submission process straightforward and benchmark easily extendable to new models and datasets, guaranteed by accessible experimental infrastructure and a unified submission procedure. Our approach makes it effortless to test and compare models in a reliable and transparent way with the possibility of quickly entering new data.

**Task diversity**   The value of the benchmark depends directly on the chosen tasks and their diversity. If an unrepresentative collection of data and tasks is used to create a benchmark, the evaluation is of limited informative value for the further development of language models. If a benchmark consists only of closely related datasets, we can evaluate only a narrow part of the model's capabilities. Hence, one of the first and the most critical tasks for us was to gather many diverse tasks for Polish that cover different domains and tasks. We wanted to cover also many sources of text data in our benchmarking environment. The model's performance could differ for books, social media, and other domain texts. Thus, having a representative collection of text data allows for evaluating the models in terms of their in-domain and out-of-domain generalization abilities. However, since Polish is a low-resource language, our choice was limited and we ended up with text classification, natural language inference, and sequence labeling tasks. We considered a few datasets for the regression task (e.g., CDSC-R or Allegro Reviews from KLEJ). However, in our opinion, mentioned datasets are unsuitable for benchmarking. It appears that annotation consistency is much lower than claimed in the original papers (Rybak et al., 2020; Wróblewska and Krasnowska-Kieraś, 2017). However, we hope to extend the benchmark with other than the mentioned above regression task in the future.

**Dataset selection**   To select proper datasets for our benchmark, we first looked at the datasets available in KLEJ (Rybak et al., 2020). Many datasets have been described in their research papers, but they were still quite hard to obtain, and of course, they were in different formats. We also noticed that some tasks defined in the KLEJ benchmark were ill-defined, e.g., the NER task as text classification instead of sequence labeling. We set out to fix these problems and widen the scope of covered tasks and domains. Actual sequence tagging datasets (KPWr-NER and NKJP POS tagging) were added to the benchmark. Regarding the classification task, we added aspect-based sentiment analysis, political advertising detection, and punctuation restoration datasets to cover more diversified tasks. We also prepared two new datasets concerning legal text (Political Abusive Clauses) and dialogue systems (DiaBiz.Kom). It is important to mention that almost all of the datasets chosen by us (i.e., KPWr-NER, AspectEmo, PolEmo, DiaBiz.Kom, PAC, Political Advertising, and PSC) have been created by researchers in the CLARIN-PL project; hence the annotation processes and inter-annotator agreements are properly described in the relevant papers. The complete list of datasets with a short description can be found in Section 3.2.

We challenged an interesting problem when extending the collection of datasets covered in the benchmark, namely, should we add a dataset that is not free and publicly available? It is an important choice when designing the benchmark. On the one hand, it contradicts the guidelines of open science, but on the other hand, it makes the benchmark more challenging and practically useful. After a lot of deliberations, we have decided to add to the benchmark the Dialogue Acts dataset — DiaBiz.Kom (more in Section 3.2) which is available only for internal usage of CLARIN-PL-Biz associates. Still, the dataset covers a significant collection of infrequent domain data for Polish targeted at spoken

language understanding. The results of modern language models for this dataset present much room for improvement — see Table 3.

**Model selection**   The next step was to select initial models in the benchmark to provide a baseline and allow for easy comparison with the already published models. Each baseline model had to be available in the HuggingFace repository, and it could not be too big since we were again limited by the amount of available compute.[3] We first took the models of different sizes provided in KLEJ – HerBERT-large is a top-1 model in the KLEJ benchmark. We used these models and our hyperparameter search module to validate the experimental setup and generate the first results for benchmark. We then utilized another popular mono-lingual encoder model and tested it against the previous one. Finally, to provide some diversity, we took the sentence model (which in fact is a cross-lingual) and ran the experiment. We plan to add new models to the benchmark to allow comparison of cross- vs. single-lingual and sentence- vs. word encoders. Table 2 shows the final collection of models used for experiments.

**Choice of metrics**   Every benchmark has to provide task-specific evaluation metrics. Even though we can focus on a single metric for a specific dataset and task in most cases, it could not be enough for many scenarios. A single metric can be insufficient to capture the varying cost of errors in many tasks. For instance, in sentiment analysis, the misclassification of "strong negative" as "negative" is far less consequential than the misclassification of the same case as "positive". In general, instead of focusing on a single metric, we want to evaluate language models using multiple metrics (and allow for the construction of compound weighted metrics). LEPISZCZE calculates many different metrics, namely, F1-score, recall, precision, and accuracy. We allow sorting submissions based on various metrics. We believe that comparing models between tasks using different metrics may be misleading, which is why we use homogeneous metrics to compare and score the models.

Many benchmarks concentrate only on metrics that evaluate the quality of predictions, without considering the cost of prediction. We believe that the omission of externalities, such as the computational requirements of models, leads to very biased rankings. The costs incurred by modern language models (i.e. in terms of their carbon footprint) can be significant and should be included in the evaluation. We track all Python environment information and completion times of all experiments for the benchmark. This way, we can build a custom leaderboard that incorporates computational costs as well. As (Ethayarajh and Jurafsky, 2020) said, a highly inefficient model would provide less utility to practitioners but not to a leaderboard since it is a cost that only the former must bear.

**Experimental environment**   We use several tools in our experimental environment to facilitate the use of the benchmark. Benchmarking involves running many experiments and tracking their performance. We actively utilize the *HuggingFace* repository to make the process of adding and testing datasets and models convenient. We unified all datasets into one accessible and easy-to-process data format, uploaded them to the HuggingFace Datasets [4] repository and ran all experiments using the HuggingFace hub. Technically, our benchmark allows us to choose any dataset or collect utterly new data, prepare data-loading scripts compatible with the HuggingFace platform, and evaluate models in the target language. We also develop our library `clarinpl-embeddings` to unify the whole process of training, validating, performing hyperparameter search, testing, and submitting results to the leaderboard almost automatically, in only a few lines of code. We believe that this step is crucial to allow continuous benchmarking and encourage researchers to submit their models or datasets. Integrating benchmark libraries with HuggingFace Datasets platform opens new possibilities to evaluate language models in a multilingual zero-shot setting for any low-resource language. We believe using a unified dataset inventory will contribute to the sustainable development of reliable evaluation data. The `clarinpl-embeddings` library is built on *PyTorch, PyTorch Lightning* and *Transformers* is easily extendable and modifiable; we plan to keep developing and expanding it over time. While the KLEJ benchmark required only a .zip file submission with predictions, we provide a great level of technical support to the user with the extensive experimental environment that reinforces the reproducibility.

---

[3] Maximum size of the model to perform hyperparameter search in a reasonable time was 350 million parameters

[4] `https://huggingface.co/datasets`

**Standard splits problems**  Many benchmarks, such as GLUE and derived works, do not reveal test sets on which the benchmarking platform calculates the final results. Using static data splits leads to over-fitting and results in quick benchmark saturation. An alternative approach is based on multiple splits(Gorman and Bedrick, 2020), allowing for evaluation of the model's performance based on many different data partitions. In our evaluation pipelines, we follow this methodology and use not only a single train, dev, and test split but multiple splits. In our benchmark, we decided to implement a new experiment to evaluate non-original splits in the next benchmark version.

**Continuous benchmarking**  The disadvantage of multi-task benchmarks such as GLUE, Super-GLUE, KLEJ, etc., is their lack of dynamics. The static benchmarks become quickly outdated and, therefore, useless from a practical perspective. As part of the CLARIN-PL-Biz [5], we plan to add more datasets, tasks, and models and maintain LEPISZCZE benchmark continuously. We encourage other associated researchers to publish datasets and models in our benchmark. Many LEPISZCZE datasets have been added with their source and author contributions. We track all model parameters and versions of each dataset. Hence, we can create a leaderboard for a specific version of the dataset in our benchmark.

## 3.2  Datasets in the benchmark

In this section, we briefly describe the final collection of datasets selected for the initial version of our benchmark. As of now, we also preserved the original splits of these datasets.

**PAC — Polish Abusive Clauses Dataset**  "I have read and agree to the terms and conditions" is one of the biggest lies on the Internet. Consumers rarely read the contracts they are required to accept. On the Internet, we probably skip most of the Terms and Conditions. However, we must remember that we have concluded many more contracts. European consumer law aims to prevent businesses from using so-called "unfair contractual terms" in their unilaterally drafted contracts, requiring consumers to accept. The PAC aims to detect "unfair contractual term" as the equivalent of an abusive clause. The task was formulated as binary text classification. The dataset has been created with the Office of Competition and Consumer Protection. This dataset uses more than 700 contracts and gathers 4,129 examples of abusive clauses and 5,127 non-abusive contract fragments.

It is worth noticing that the PAC dataset is important from an ethical point of view. Particularly it is based on actual agreements. The Office of Competition and Consumer Protection employees have checked the dataset to see if it contains Personal Identifiable Information (PII). A couple of such examples have been removed from the texts.

**AspectEmo**  Corpus (Kocoń et al., 2021) is an extended version of the publicly available PolEmo 2.0 corpus. The AspectEmo corpus consists of 1,465 online customer reviews from the following domains: school, medicine, hotels, and products. All documents are annotated at the aspect level with six sentiment categories: strong negative, weak negative, neutral, weakly positive, and strong positive.

**CDSC-E**  Compositional Distributional Semantics Corpus (Wróblewska and Krasnowska-Kieraś, 2017) is an entailment classification task. It consists of 1000 pairs of sentences and human-annotated entailment labels for each pair. There are three possible classes: entailment, contradiction, and neutral.

**Dialogue Acts — DiaBiz.Kom**  It consists of 1,277,962 tokens in 1,104 transcribed call center phone conversations spanning eight domains. Each example is annotated by three linguists (in a 2+1 system, with an inter-annotator agreement of Positive Specific Agreement equal to 0.78 for annotation borders and categories and 0.86 for annotation borders) with dialogue acts in compliance with ISO 64217-2:2012 standard with layer of information concerning communicative functions. Within the benchmark, we consider the task of dialogue act classification, where each utterance is provided with its role in the dialogue. DiaBiz.Kom is annotation layer on top of DiaBiz (Pęzik et al., 2022) — corpus of Polish call center dialogues.

---

[5] `http://clarin-pl.eu/index.php/en/home/`

**DYK**   Did You Know (pol. Czy wiesz?) is a dataset that consists of 4,721 human-annotated question-answer pairs. It was simplified by (Rybak et al., 2020) to binary classification to label denoting if the answer contained in the Wikipedia article is factually correct in light of the stated question.

**KPWr-NER**   is a part of the Polish Corpus of Wrocław University of Technology (Korpus Języka Polskiego Politechniki Wrocławskiej) (Broda et al., 2012) is a named entity recognition dataset focusing on fine-grained categories of entities (82 classes) using BIO notation. It contains 13,959 training and 4,323 test human annotated sentences, originating from texts covering a large variety of domains, genres, and sources.

**NKJP-POS**   is a part of the National Corpus of Polish (Narodowy Korpus Języka Polskiego) (Przepiórkowski et al., 2012). Its objective is the part-of-speech tagging task. The dataset contains 85,663 sentences tagged with 35 tags. During the creation of the corpus, texts were annotated by humans from various sources, covering many domains and genres.

**PolEmo 2.0**   (Kocoń et al., 2019) is a dataset of online consumer reviews from four domains: medicine, hotels, products, and university. It consists of 8,216 reviews having 57,466 sentences. The aim is to predict one of the sentiment classes: positive, negative, neutral, or ambiguous. During the development of the KLEJ benchmark (Rybak et al., 2020) two tasks that differ in the context used during evaluation have been created: **in-domain** and **out-domain**. In contrast, we preserved the original data split and utilized all domains.

**Political Advertising**   dataset (Augustyniak et al., 2020) aims for detecting specific text chunks and categories of political advertising in the Polish language. It contains 1,705 human-annotated tweets tagged with nine categories, constituting campaigning under Polish electoral law. The authors achieved 0.65 inter-annotator agreement (Cohen's kappa score) for the sequence labeling task, and they used an additional annotator to resolve the mismatches between the first two annotators, improving the final consistency of annotations.

**PSC**   Polish Summaries Corpus (Ogrodniczuk and Kopeć, 2014) consists of 569 news summaries done by human annotators. We used the simplified version developed by (Rybak et al., 2020) for the purpose of the KLEJ benchmark. They formulated a binary paraphrase classification task by matching positive and negative pairs using the procedure detailed in the publication.

**Punctuation Restoration**   is a crowd-sourced text and audio dataset of Polish Wikipedia pages read out loud by Polish lectors. The base dataset is divided into conversational (WikiTalks) and information (WikiNews) parts. Then the texts were read by hundred people, which resulted in 36 hours of transcription. Punctuation restoration includes 1000 texts - 800 trains and 200 test examples. This dataset is part of PolEval 2021 Competition [6].

## 4   Experiments

We conducted experiments on 13 datasets and five language models. Each language model was fine-tuned for a given dataset and was evaluated separately. Experiments were performed with our developed library `clarinpl-embeddings`, which provides predefined pipelines for text classification, text pairs classification, and sequence labeling. To ensure reproducibility of our experiments, we utilized MLOps tools: Data Version Control (DVC) (Kuprieiev et al., 2022) for pipelines and tracking of datasets and models; Weight&Biases (Biewald, 2020) for experiments summaries and metrics tracking. We share the code of our experiments on GitHub repository [7] and model tracking dashboard [8].

---

[6]`http://2021.poleval.pl/tasks/`
[7]`https://github.com/CLARIN-PL/LEPISZCZE`
[8]`https://wandb.ai/embeddings/LEPISZCZE`

Table 1: Datasets available in the LEPISZCZE benchmark with sizes of the train, dev, and test sets. The datasets that were previously incorporated into the KLEJ benchmark are marked with * symbol. **WIP** denotes the dataset for which we present preliminary results.

| Name | Domain | Task | Train | Dev | Test | #Classes |
|---|---|---|---|---|---|---|
| CDSC-E* | image captions | Entailment Classification | 8000 | 1000 | 1000 | 3 |
| DYK* | Wikipedia | Q&A Classification | 4154 | 0 | 1029 | 2 |
| PolEmo 2.0 (In-Domain)* | online reviews | Sentiment Analysis | 5783 | 723 | 722 | 4 |
| PolEmo 2.0 (Out-Domain)* | online reviews | Sentiment Analysis | 5783 | 494 | 494 | 4 |
| PSC* | news | Paraphrase Classification | 4302 | 0 | 1078 | 4 |
| Abusive Clauses | legal texts | Abusive Clauses Detection | 4284 | 1519 | 3453 | 2 |
| AspectEmo | online reviews | Aspect-based Sentiment Analysis | 1173 | 0 | 292 | 7 |
| KPWr NER | misc. | NER | 13959 | 0 | 4323 | 82 |
| NKJP POS | misc. | POS Tagging | 78219 | 0 | 7444 | 35 |
| PolEmo 2.0 | online reviews | Sentiment Analysis | 6573 | 823 | 820 | 4 |
| Political Advertising | social media | Political Advertising Detection | 1020 | 340 | 341 | 9 |
| Punctuation Restoration | Wikipedia Talk, Wikinews | Punctuation Restoration | 800 | 0 | 200 | 8 |
| Dialogue Acts (WIP) | call center phone conversations | Dialogue Acts Classification | 70454 | 8807 | 8807 | 54 |

## 4.1 Initial models for benchmark

We picked four recent transformer-based language models for Polish publicly available in the HuggingFace hub, along with one multilingual XLM-RoBERTa model. We present those models with their total number of parameters and repository location in Table 2. For fine-tuning, we utilize sequence and token classification models from the transformers library (Wolf et al., 2020), consisting of a single linear classification layer with dropout. For the initial datasets evaluation, we chose both cased and uncased versions of the language models.

Table 2: Language Models used for experiments. All models can be accessed via the HuggingFace repository.

| Model | #Params | HuggingFace Repository Name |
|---|---|---|
| PolBERT (base, cased), (Kłeczek, 2020) | 132M | dkleczek/bert-base-polish-cased-v1 |
| PolBERT (base, uncased), (Kłeczek, 2020) | 132M | dkleczek/bert-base-polish-uncased-v1 |
| HerBERT (base, cased) (Mroczkowski et al., 2021) | 124M | allegro/herbert-base-cased |
| HerBERT (large, cased) (Mroczkowski et al., 2021) | 355M | allegro/herbert-large-cased |
| XLM-RoBERTa (paraphrase) (Reimers and Gurevych, 2019) | 278M | sentence-transformers/paraphrase-xlm-r-multilingual-v1 |

## 4.2 Hyper-parameter search (HPS)

To fairly compare different transformer models across various tasks, we performed a hyper-parameter search (HPS) to obtain the best configuration for fine-tuning the language model to a particular task. We performed a hyper-parameter search separately for each combination of tasks and language models (which we restricted to 100 iterations). Under the hood, we utilized the Optuna framework (Akiba et al., 2019) wrapper from the `clarinpl-embeddings` library. We also logged each run in the hyper-parameter search via Weights&Biases PyTorch Lightning logger.

Experiments were computed using a server with five Titan RTX GPU cards. We logged over 6000 runs in the Weight&Biases dashboard, which took over 2000 hours to complete. We reported metrics, hyper-parameters, dataset information, and package versions in each run.

We used macro averaged F1 measure as the metric for the objective function. Evaluation of models in the hyper-parameter search stage was performed on the validation subset of the dataset. In case validation subset was missing, we randomly sampled 10% of the training subset. After obtaining the best hyper-parameter configuration, we no longer need such a subset, so we use original subsets for the final model evaluation.

For each dataset and language model pair, we choose the best configuration from the HPS process in terms of the best F1-macro score on the validation subset. We retrain models five times and calculate various metrics on test sets such as accuracy, precision, recall, and F1 with different averaging (micro, macro, weighted) and class or tags metrics (accuracy, precision, recall, and F1).

### 4.3 Results of an initial set of trained models

Evaluation results are presented in Table 3, where we report macro averaged F1 metric for each dataset. Other metrics results can be found in the appendix Section A.3. As we observe the performance of models above 80% in text classification datasets (except out-domain dataset), these models perform poorly considering most sequence tagging tasks. Even the best performing model (HerBERT, large) shows F1-macro around 39% for AspectEmo and 46% in Punctuation Restoration. Considering those results, we can state that we still need better models that can cope with complex and under-represented tasks. Multilingual models XLM-RoBERTa and HerBERT (large) are comparable only for one dataset (around 2 percentage points difference for the Abusive Clauses dataset). However, for other tasks, the gap is much bigger, and it fails even more in sequence tagging tasks with a difference of up to 70%.

We also reported preliminary results for Dialogue Acts classification tasks. For the tested language model, we achieved a comparable performance of about 50%. Considering limited computational resources and fine-tuning schemes, these results may be updated in our future work.

Table 3: Macro F1 performance of evaluated models on the test subsets. We present values as the mean and standard deviations over 5 model retrains. The mean rank row is the average of a ranking established on the mean of model retrains. Values marked with **Bold** present the best results for a single dataset. Additionally, we indicate datasets previously appeared in the KLEJ benchmark with *. **WIP** denotes the dataset for which we present preliminary results.

| | HerBERT (base, cased) | HerBERT (large, cased) | PolBERT (base, cased) | PolBERT (base, uncased) | XLM-RoBERTa (paraphrase) |
|---|---|---|---|---|---|
| CDSC-E* | **90.96 ± 0.73** | 90.48 ± 0.20 | 88.95 ± 0.31 | 90.62 ± 0.27 | 82.62 ± 0.88 |
| DYK* | **82.39 ± 1.43** | 79.58 ± 0.59 | 75.87 ± 0.98 | 74.41 ± 1.15 | 58.93 ± 7.98 |
| PolEmo 2.0 In-Domain* | 88.10 ± 0.36 | **88.34 ± 0.63** | 85.32 ± 0.45 | 85.71 ± 0.40 | 83.75 ± 0.45 |
| PolEmo 2.0 Out-Domain* | **57.31 ± 2.93** | 57.08 ± 2.03 | 54.10 ± 3.82 | 54.29 ± 1.83 | 45.12 ± 3.40 |
| PSC* | 97.90 ± 0.24 | 98.33 ± 0.69 | **98.95 ± 0.13** | 98.87 ± 0.10 | 58.85 ± 1.49 |
| Abusive Clauses | 85.66 ± 0.58 | **86.57 ± 0.91** | 85.93 ± 0.66 | 85.74 ± 0.86 | 84.32 ± 0.71 |
| AspectEmo | 37.28 ± 0.71 | **39.44 ± 1.74** | 30.01 ± 0.58 | 31.48 ± 1.06 | 18.42 ± 0.98 |
| KPWr NER | **54.22 ± 0.76** | 52.68 ± 1.39 | 48.01 ± 0.76 | 40.21 ± 0.50 | 36.13 ± 0.44 |
| NKJP POS | 94.59 ± 0.56 | **96.14 ± 0.38** | 94.34 ± 0.61 | 94.54 ± 0.19 | 90.29 ± 0.51 |
| PolEmo 2.0 | 86.78 ± 0.79 | **89.33 ± 0.49** | 85.89 ± 1.25 | 85.83 ± 0.47 | 84.12 ± 0.47 |
| Political Advertising | 61.42 ± 1.38 | 62.16 ± 0.14 | 58.94 ± 1.92 | **62.52 ± 1.23** | 56.68 ± 0.94 |
| Punctuation Restoration | 45.59 ± 0.38 | **46.68 ± 0.61** | 38.89 ± 0.91 | 41.31 ± 0.59 | 14.33 ± 1.94 |
| Dialogue Acts (WIP) | 49.54 ± 0.74 | **51.11 ± 0.85** | 50.20 ± 1.32 | 48.87 ± 0.90 | 49.05 ± 0.39 |
| Mean rank | 2.15 | 1.62 | 3.23 | 3.08 | 4.92 |

# 5   Limitations

This study has potential limitations which are listed below. First, we do not give a human baseline score to datasets in the benchmark as in SuperGLUE (Wang et al., 2019). Second, in the initial version of the benchmark, we do not solve the standard split problem and evaluate the model on predefined original splits, we work on the second batch of results with more focus on diversified splits. Finally, due to practical constraints, the initial version of the benchmark does not include baselines with static embeddings, but they will be added as well in the second batch of results. Our intention is to keep the benchmark dynamic; we plan to add new datasets with various tasks which were not covered in this version of the benchmark, including NLG tasks such as summarization or translation.

# 6   Conclusions and Future Work

In this paper, we have introduced LEPISZCZE , a new comprehensive benchmark for Polish NLP. LEPISZCZE is characterized by the large variety of NLP tasks and high-quality operationalization of the benchmark. The benchmarking approach is designed to maximize the flexibility and portability of other low-resourced languages. Adding new models, datasets, or NLP tasks is simple and intuitive. The benchmark internally supports data versioning and model tracking for improved reproducibility. In the first run of the benchmark, we tested 13 experiments (task and dataset pairs) based on the five most recent LMs for Polish to prove the usability and usefulness of LEPISZCZE .

An important added value of the paper is sharing our experiences collected during the work on the benchmark. We hope that NLP researchers working on other low-resourced languages will find our comments and suggestions useful. Below we summarize the most important issues encountered during our work on LEPISZCZE :

- *multiple metrics*: it is important to provide implementations of multiple metrics that can be measured and compared across NLP tasks and datasets; evaluating language models on a single metric produces a distorted view of models' capabilities,

- *diversity trumps openness*: one should not refrain from including closed datasets in the evaluation; they prevent over-fitting (as LMs are unlikely to see these datasets during training) and provide a good estimation of LM's performance on difficult tasks,

- *include prediction costs*: the quality of prediction as measured by traditional NLP metrics is not enough, for the benchmark to be practically useful, one must be able to compare the computational resources consumed by LMs as well,

- *interface matters*: making the benchmark interface simple and conventional hugely improves its usefulness and the probability of wide adoption; in the space of NLP models, the `HuggingFace` is the obvious choice of interface blueprints.

We plan to add more Natural Language Understanding and Spoken Language Understanding tasks to the benchmark. We want to use our `clarinpl-embeddings` library to evaluate language models, other contextual embeddings, and static word representations with simpler than transformer-based models. The critical direction in the benchmark is to run experiments not only with the predefined data splits (Gorman and Bedrick, 2020) but also to use other splits to check the model's robustness properly. We hope that this benchmark will encourage the scientific community to work in transparent and reproducible environments, leading to a rapid improvement of the current language technology for the Polish language.

## Acknowledgments and Disclosure of Funding

The work was partially supported by (1) the Polish Ministry of Education and Science, CLARIN-PL; (2) the European Regional Development Fund as a part of the 2014-2020 Smart Growth Operational Programme, CLARIN – Common Language Resources and Technology Infrastructure, (3) project CLARIN-Q (agreement no. 2022/WK/09), and (4) the Department of Artificial Intelligence at Wroclaw University of Science and Technology.

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
