# OpenReview forum: "This is the way: designing and compiling LEPISZCZE, a comprehensive NLP benchmark for Polish"
_NeurIPS.cc/2022/Track/Datasets_and_Benchmarks — NeurIPS 2022 Datasets and Benchmarks _

### Official Review · Reviewer_5XGN · 2022-07-24
**This paper benchmarked existing language models (either pretrained on Polish or multilingually) on a variety of NLP tasks in Polish**

**Rating:** 6
**Confidence:** 4
**Correctness:** The claims made in the paper are corr…
**Clarity:** 1) PAC dataset description is not cle…

**Strengths:**

1) multiple splits are considered to evaluate performance more comprehensively
2) a variety of NLU tasks for Polish are considered and evaluated in this paper

**Weaknesses:**

1) The contribution to datasets is limited since all of the used datasets are not new and have been investigated in other literature before.
2) From the perspective of benchmarking, most of the studied models are bert-based. However, there are quite a few recent multilingual LMs and the comparison would be more complete if these models are considered, e.g., Roberta (sdadas/polish-roberta-base-v2), mT5 from Google (google/mt5-base), InfoXLM from Microsoft (microsoft/infoxlm-base), XGLM from meta (facebook/xglm-564M), mGPT from SberDevices (sberbank-ai/mGPT), etc.
3) Since the paper is aimed at benchmarking NLP tasks for Polish (as paper title suggests), more important NLP tasks besides text/token classification could be considered, e.g., generation, question answering, summarization, translation, etc.

**Additional Feedback:**

I have a few related questions and hope they could be answered by the authors:

1) It has been mentioned that “Using static data splits leads to over-fitting and results in quick benchmark saturation. ", as the motivation for multiple split evaluation. But for some of the benchmarks, the ground-truth of test splits is unprovided and we can only get performance by submitting predictions. Why this kind of benchmark still face the problem of "over-fitting" and "quick saturation"?
2) The hyperparameter names and ranges are not provided in Section 4.2. It would be great if a table reporting those details is included in the main paper or appendix
3) Do you mind adding the resource and year of numbers for Figure 1? Since the number of speakers for a specific language changes year by year, some values I searched online does not align well with those demonstrated in the figure, e.g.,
- English speaker is around: 1350 million in 2021 (https://www.babbel.com/en/magazine/how-many-people-speak-english-and-where-is-it-spoken#:~:text=Out%20of%20the%20world's%20approximately,%2C%20in%20third%2C%20comes%20English.)
- Chinese speaker is around: 1300 million in 2021 (https://www.babbel.com/en/magazine/the-10-most-spoken-languages-in-the-world#:~:text=1.,spoken%20language%20in%20the%20world.)
- Indian speaker is around: 615 million in 2019 (https://newsonair.gov.in/News?title=Hindi-is-3rd-most-spoken-language-in-the-world-with-615-million-speakers-after-English%2C-Mandarin&id=381514#:~:text=Hindi%20is%20the%203rd%20most,list%20with%201%2C132%20million%20speakers.)


Below are some typos and could be corrected for the next version of this paper:
1) should be "reveal" rather than "revile" (line 166 of page 5)
2) "started working" rather than "started work" (line 183 of page 5)
3) left double quote should be “  rather than  ”​ (multiple occurrences between line 188 and line 200 for PAC dataset description)


**Documentation:**

The paper has provided links to the codes, experiment logs and datasets in Huggingface.

However, the dashboard (wandb logs) is unclear to rank different methods across datasets. Documents to report necessary results and major conclusions are missing.


**Ethics:**

The paper has discussed ethical concerns when introducing the studied datasets.

**Relation To Prior Work:**

The paper has mentioned this work as an extension of the existing benchmark for Polish NLP KLEJ. However, there are a few points unclear related to KLEJ and LEPISZCZE:
1) Why regression tasks studied in KLEJ are not studied here: CDSC-R and AR
2) Why CBD (Cyberbullying detection) and NKJP-NER (NER classification) are not included in this work?

**Summary And Contributions:**

Summary:
This paper introduces the benchmark LEPISZCZE to benchmark for Polish NLP. Specifically, results on 13 tasks from give LMs for Polish are reported in this paper.

Contributions:
1) more dataset in Polish are included to evaluate Polish LMs
2) design the benchmark from the perspectives of data splits, continuous maintainess, etc
3) lessons learned during benchmark design are summarized and are expected to be helpful for benchmarks concerning other low-resource languages

---

> ### Author Response · Authors · 2022-08-25
> **Authors' response to Reviewer 5XGN (Part I)**
>
> **Q1**: The contribution to datasets is limited since all of the used datasets are not new and have been investigated in other literature before.
> **A1**: We want to emphasize that two new datasets, namely PAC (Polish Abusive Clauses) and Dialogue Acts (Diabiz.Kom), have not been investigated in the literature yet. Moreover, we do not limit the benchmark to the datasets we presented in the current version, and we aim to add more datasets in the future.
>
> ---
> **Q2**: From the perspective of benchmarking, most of the studied models are bert-based. However, there are quite a few recent multilingual LMs and the comparison would be more complete if these models are considered, e.g., Roberta (sdadas/polish-roberta-base-v2), mT5 from Google (google/mt5-base), InfoXLM from Microsoft (microsoft/infoxlm-base), XGLM from meta (facebook/xglm-564M), mGPT from SberDevices (sberbank-ai/mGPT), etc.
> **A2**: We are aware that a broader range of language models exists compared to the ones we included in the benchmark. We focused on providing solid baselines in the first iteration of our analysis. Furthermore, we gave the users an option to create a submission to the benchmark, according to the instructions provided on the leaderboard page [https://lepiszcze.ml/submission/](https://lepiszcze.ml/submission/). Moreover, we encountered hardware limitation issues due to GPU memory limits and time-intensive computation.
> ---
> **Q3:** Since the paper is aimed at benchmarking NLP tasks for Polish (as paper title suggests), more important NLP tasks besides text/token classification could be considered, e.g., generation, question answering, summarization, translation, etc.
> **A3**: A more diverse range of tasks beyond classification is the goal for the next version of the benchmark. However, this is not only limited to adding new tasks in the library but often goes along with creating a dataset tailored for that task. In the Polish language, we often have to cope with a situation where there are no publicly available datasets (or some of them are non-representative and have a limited size). In the nearest future, we will extend the benchmark with dialogue-related tasks such as question answering, slot filling, or answer generation.
>
> ---
> **Q4**: Clarity
>
> - PAC dataset description is not clear, mixing examples and dataset descriptions
>
> **A:** We simplified the PAC dataset description by reducing examples and highlighting the essential information. Moreover, the extended version is published on HuggingFace Hub ([https://huggingface.co/datasets/laugustyniak/abusive-clauses-pl](https://huggingface.co/datasets/laugustyniak/abusive-clauses-pl)) and the LEPISZCZE website [https://lepiszcze.ml/datasets/](https://lepiszcze.ml/datasets/)
> - In line 172 of page 5: what does “other splits” mean here?
>
> **A: “**Other splits” means non-original ones that were not proposed by the authors of the dataset. However  we will tackle this issue the next version of our benchmark (added in “Limitiations” section as well)
> - From line 210 to 219 on page 6: details especially information links could be provided as footnotes. Otherwise, it becomes difficult to distinguish websites and normal texts.
>
> **A:** For clarification, we moved this information to descriptions on the HuggingFace hub and LEPISZCZE leaderboard site.
> - Meanwhile, there is an undefined citation (question mark) for the corpus of polish call center dialogs.
>
> **A**: We have fixed missing citation.
> - From line 244 to 247 on page 6: is this a summarization task or classification task? Since the original dataset is built for summarization and previous work formalized it as text-similarity, how does this paper define the task on this dataset? According to Table 1, PSC is defined as a paraphrase classification task. It could be much more clear if the dataset description part could declare more straightforwardly.
>
> **A**: We have improved the description to be more straightforward; thus, it no longer suggests that this is a summarization task, but a simplified version, i.e. a binary paraphrase recognition task.
>
> ---
> **Q5**. Considering the variety of datasets, the description could be more clear if the datasets are listed according to their task type, following what has been done in glue paper (single-sentence tasks, similarity and paraphrase tasks, inference tasks)
>
> **A5**: We did not use such task categorisation in the first version of the benchmark. Nevertheless, in the next version of the benchmark, we plan to cover more diversified tasks so that we may introduce similar categorisation.

---

> > ### Author Response · Authors · 2022-08-25
> > **Authors' response to Reviewer 5XGN (Part II)**
> >
> > **Q6: Why regression tasks studied in KLEJ are not studied here: CDSC-R and AR**
> >
> > **A6:** We considered a few datasets for the regression task. However, in our opinion, text regression datasets in KLEJ are unsuitable for benchmarking.
> >
> > - According to the original paper that presented CDSC-R, the annotation was done by averaging rates of human annotators that assessed relatedness on a Likert scale (0-5). Such an assumption is very subjective, so the Fleiss kappa score was low at 0.337.
> > - Allegro Reviews consists of reviews collected from an e-commerce platform. Manual analysis of the samples showed that a lot of text reviews do not affect the score (1-5) clearly.
> >
> > We hope to extend the benchmark by regression task in the future.
> >
> > ---
> >
> > **Q7:** Why CBD (Cyberbullying detection) and NKJP-NER (NER classification) are not included in this work?
> >
> > **A7:**
> >
> > - CBD - In the near future, we plan to add this dataset to our benchmark. For now, we excluded this dataset due to limited computational resources - we could not provide a complete experimental analysis with an extensive hyperparameter search.
> > - NKJP-NER is a simplified version of the typical NER task as it is defined as document classification in comparison to the standard sequence labeling task. Moreover, the dataset was prepared on the basis of a simplified typology of named entities. We provided the KPWR-NER dataset, which is appropriately defined as a sequence labeling task with fine-grained NE categories.
> >
> > ---
> >
> > **Q8**: However, the dashboard (wandb logs) is unclear to rank different methods across datasets.
> >
> > **A8**: We added a leaderboard page [https://lepiszcze.ml](https://lepiszcze.ml/) for easier ranking and accessing metrics. Wandb logs can still be utilized to get more insights into the model’s training.
> >
> > ---
> >
> > **Q9**: Documents to report necessary results and major conclusions are missing.
> >
> > **A9**: We added more results in the Appendix section and added more descriptions of results and averaged rankings for models.
> >
> > ---
> >
> > **Q10**: It has been mentioned that “Using static data splits leads to over-fitting and results in quick benchmark saturation. ", as the motivation for multiple split evaluation. But for some of the benchmarks, the ground-truth of test splits is unprovided and we can only get performance by submitting predictions. Why this kind of benchmark still face the problem of "over-fitting" and "quick saturation"?
> >
> > **A10**: We can submit multiple predictions and then find out how the metric in the benchmark is computed. It is the same problem as in Kaggle competitions. Moreover, for most of the datasets we used in our benchmark, the data is already available hence it could be easy to find proper predictions and insert them into submissions. We decided not to hide test data and be completely transparent. In addition, we work on adding multiple different split evaluations to the leaderboard.
> >
> > ---
> > **Q11**: The hyperparameter names and ranges are not provided in Section 4.2. It would be great if a table reporting those details is included in the main paper or appendix
> >
> > **A11**: We added new details on how hyperparameter search space was defined (see our Appendix). Moreover, best configuration of parameters is detailed in the leaderboard view of the submission (via clicking the “i” button on the page” e.g., [https://lepiszcze.ml/by-dataset/](https://lepiszcze.ml/by-dataset/))
> >
> > ---
> >
> > **Q12**: Do you mind adding the resource and year of numbers for Figure 1? Since the number of speakers for a specific language changes year by year, some values I searched online does not align well with those demonstrated in the figure,
> >
> > **A12**: We added missing citations with the year of publication to speakers per language statistics.
> >
> > ---
> >
> > **Q13**: Below are some typos and could be corrected for the next version of this paper
> >
> > **A13**: We fixed all mentioned typos.

---

### Official Review · Reviewer_FCyK · 2022-07-25
**A good benchmark for Polish NLP**

**Rating:** 6
**Confidence:** 3

**Strengths:**

- The benchmark construction and improvement for Polish NLP is well motivated. And the discussion of limitations of previous benchmarking seems reasonable to this reviewer.
- Codes, models, datasets, and parameters are clearly provided, which promotes reproducibility.


**Weaknesses:**

- The benchmark construction process is unclear:
  - The selection criteria of the five datasets from Klej and new datasets are not clarified. Moreover, what is the overall design idea behind all the datasets/tasks in LEPISZCZE?
   - The evaluation metrics for different datasets are not described.
   - The organization of different datasets. In the paper (line 186), the authors claim that they unify different datasets into the same form. How the authors do that is not described.
    - The splits of train, dev, and test sets are not clear. Do the authors use the original splits of the corresponding datasets?

- The limitations of previous benchmarking are not addressed. Given the existing benchmark Klej, the paper does not show much improvement besides adding new datasets:
    - Single metric evaluation is identified as a limitation of prior work. But the paper seems to also use a single metric for model evaluation.
    - The paper does not show improvements over standard split problems.
    - The paper does not show sufficient proof of ensuring continuous benchmarking.
    - The benchmark datasets seem to all relate to classification tasks, other important tasks such as generation tasks (e.g., summarization, translation) are not included.
    - According to the evaluation, HerBERT seems a clear winner. It outperforms other models on the majority of newly proposed tasks, which weakens benchmark contributions to some extent compared to the situations where different models perform better at different tasks (in Klej).

**Additional Feedback:**

I did not see the supplementary material in this submission as mentioned in the paper (Line 89,98). More importantly, I did not see the leaderboard (Line 111, 117) as promised in the paper.

**Clarity:**

The paper is a bit hard to follow, especially since the selection criteria of datasets in the benchmark are not clarified. And the paper can reduce the length of background/motivation about benchmarking Polish NLP.

**Correctness:**

As mentioned in the weaknesses, the paper does not clarify the benchmark construction process. Moreover, the paper claims that the proposed benchmark uses different metrics while it does not report and compare them in the evaluation (it just uses the F1 score), which is confusing to this reviewer.

**Documentation:**

As a benchmark paper, I think the implementation details (e.g., configurations, datasets links, models) provided in Github are sufficient and clear for reproduction, although the reproducibility requires high effort.

**Ethics:**

No. But I am not an expert.

**Relation To Prior Work:**

No. The paper points out the limitations of Klej (an existing Polish NLP benchmark), including task formulation, and single evaluation metric. However, the paper does not show and clarify the corresponding improvements over the aforementioned limitations.
Moreover, this reviewer suggests that there should be more discussion about related work on benchmark construction (e.g., tasks design and formulation, evaluation metrics) of other languages, especially low-resourced languages.


**Summary And Contributions:**

This paper extends an existing benchmark for Polish NLP (i.e., Klej) and presents LEPISZCZE, a new benchmark that integrates five tasks from the Polish benchmark and eight new datasets. Also, the paper conducts experiments on the benchmark based on the F1 macro performance of some recent LMs for Polish.

---

> ### Author Response · Authors · 2022-08-25
> **Authors' response to Reviewer FCyK (Part I)**
>
> **Q1:** *The selection criteria of the five datasets from Klej and new datasets are not clarified. Moreover, what is the overall design idea behind all the datasets/tasks in LEPISZCZE?*
>
> **A1:** The criteria for the task, dataset, and model selection are provided in Section 3.1 of the updated version of the paper. We also introduced new datasets in the revised Section 3.2 and we elaborated more on the motivation of their inclusion in the benchmark (PAC, DiaBiz.Kom).
>
> - Regression Datasets (CDSC-R, AR):  We considered a few datasets for the regression task. However, in our opinion, text regression datasets in KLEJ are unsuitable for benchmarking. According to the original paper that presented CDSC-R, the annotation was done by averaging rates of human annotators that assessed relatedness in Likert scale (0-5). Such assumption is very subjective so the Fleiss kappa score was low 0.337. Allegro Reviews consists of reviews collected from an e-commerce platform. Manual analysis of the samples showed that a lot of text reviews do not affect the score (1-5) in a clear way. We hope to extend the benchmark by regression task in the future.
> - CBD: In the near future, we plan to add the remaining CBD dataset as well as other text-based regression datasets to our benchmark.
> - NKJP-NER is a simplified version of the typical NER task as it is defined as document classification in comparison to the standard sequence labeling task. Moreover, the dataset was prepared on the basis of a simplified typology of named entities. We provided the KPWR-NER dataset, which is appropriately defined as a sequence labeling task with fine-grained NE categories.
>
> ---
>
> **Q2**: *The evaluation metrics for different datasets are not described.*
>
> **A2**: We evaluate text classification and text pair classification datasets using scikit-learn metrics hugging face wrapper [https://github.com/CLARIN-PL/embeddings/blob/main/embeddings/evaluator/text_classification_evaluator.py](https://github.com/CLARIN-PL/embeddings/blob/main/embeddings/evaluator/text_classification_evaluator.py). For sequence labeling, we utilise seqeval library with its own wrapper for unit tag scheme (like POS tags). We used standard evaluation metrics for classification tasks, including sequence labeling (precision, recall, accuracy, F1-micro, F1-macro), which was mentioned in our paper.
>
> ---
>
> **Q3**, **Q4**: *Moreover, the paper claims that the proposed benchmark uses different metrics while it does not report and compare them in the evaluation (it just uses the F1 score), which is confusing to this reviewer.*
>
> *Single metric evaluation is identified as a limitation of prior work. But the paper seems to also use a single metric for model evaluation.*
>
> **A3**, **A4**: As in the first version of the paper, we reported only the macro F1 score. We added other results in the Appendix section. Moreover, all metrics can be easily accessed via our leaderboard page or in the logs attached directly to submissions on our leaderboard. We also added averaged class-specific metrics in the detailed view of the leaderboard.
>
> ---
>
> **Q5**: *The organization of different datasets. In the paper (line 186), the authors claim that they unify different datasets into the same form. How the authors do that is not described.*
>
> **A5**: Regarding the organization of different datasets, **we refer to lines 129-131 in the initial version of the paper, where we state that data organization is performed using the HuggingFace repository.
>
> ---
>
> **Q6**: *The splits of train, dev, and test sets are not clear. Do the authors use the original splits of the corresponding datasets?*
>
> **A6**: The original splits were preserved. In line 282 of the paper's first version, we also mention that we randomly sampled 10% of training subset in case the validation subset was missing. However, this subset was only used for the hyperparameter search phase, and we used the original splits for model evaluation.
>
> ---
>
> **Q7**: *The paper does not show improvements over standard split problems.*
>
> **A7**: As of now, we recognize the standard split problem, and we intend to address it in the next version of our benchmark.
>
> ---
>
> **Q8**: *The paper does not show sufficient proof of ensuring continuous benchmarking.*
>
> **A8**: We published leaderboard: [lepiszcze.ml](http://lepiszcze.ml/), which automatically parses submission into the results table. We additionally provided a Submission section on our leaderboard page describing how one can add new evaluation scores to the benchmark.

---

> > ### Author Response · Authors · 2022-08-25
> > **Authors' response to Reviewer FCyK (Part II)**
> >
> > **Q9**: *The benchmark datasets seem to all relate to classification tasks, other important tasks such as generation tasks (e.g., summarization, translation) are not included.*
> >
> > **A9**: We added limitation sections in which we describe this issue. Nevertheless, we currently aim to develop more technologies for conversational data and to add models for question answering, slot filling, and answer generation. For this reason, we added a new dataset - DiaBiz.Kom - which contains natural language conversations in Polish. In future work, we plan to use this new dataset as a basis for testing neural language models in text generation tasks (this will be included as a separate task on our leaderboard).
> >
> > ---
> >
> > **Q10**: *According to the evaluation, HerBERT seems a clear winner. It outperforms other models on the majority of newly proposed tasks, which weakens benchmark contributions to some extent compared to the situations where different models perform better at different tasks (in Klej).*
> >
> > **A10:** We aimed to deliver strong baselines for the benchmark. In parallel, we are working on overcoming hardware difficulties which would allow us to deliver a new range of language models dedicated to Polish language.
> >
> > ---
> >
> > **Q11**: *The paper points out the limitations of Klej (an existing Polish NLP benchmark), including task formulation, and single evaluation metric. However, the paper does not show and clarify the corresponding improvements over the aforementioned limitations. Moreover, this reviewer suggests that there should be more discussion about related work on benchmark construction (e.g., tasks design and formulation, evaluation metrics) of other languages, especially low-resourced languages.*
> >
> > **A11**:  We added the section “3.1 Benchmark designing and construction process” that addresses those issues. The limitations and our solutions are as follows:
> >
> > 1. *Limitation*: Task-specific metrics can lead to an erroneous conclusion about model performance. *Solution*: We use single, homogeneous metric evaluation for model ranking instead of task-specific, heterogeneous ones (Section 3.1, paragraph: Choice of metrics). Still, one can generate a ranking with respect to different evaluation metrics.
> > 2. *Limitation*: Not representative enough set of tasks. *Solution*: We add more diverse and difficult tasks (Section 3.1, paragraph Task diversity and Dataset selection).
> > 3. *Limitation*: Insufficient technical support, which makes the benchmark stale and difficult to maintain *Solution:* We developed our library (embeddings) to unify the whole process of training, validating, performing hyperparameter search, testing, and submitting results to the leaderboard almost automatically (just in a few lines of code). We additionally provided a Submission section ([https://lepiszcze.ml/submission/](https://lepiszcze.ml/submission/)) on our leaderboard page describing how to add new submissions to the benchmark.
> >
> > Unfortunately, we did not analyse the existing approaches to benchmark construction for other low-resource languages.
> >
> > ---
> >
> > **Q12**: *The paper is a bit hard to follow, especially since the selection criteria of datasets in the benchmark are not clarified. And the paper can reduce the length of background/motivation about benchmarking Polish NLP.*
> >
> > **A12**: We addressed these concerns in the updated version of the paper.
> >
> > ---
> >
> > **Q13**: *I did not see the supplementary material in this submission as mentioned in the paper (Line 89,98). More importantly, I did not see the leaderboard (Line 111, 117) as promised in the paper.*
> >
> > **A13**: We added supplementary material that contains a table with other NLP benchmarks considered in the related work section, hyperparameter search configuration, and tables with all evaluation metrics calculated for the datasets in the benchmark. The promised leaderboard is available at [https://lepiszcze.ml](https://lepiszcze.ml/).

---

### Official Review · Reviewer_3SWT · 2022-07-25
**A benchmark for Polish, 13 tasks and five models**

**Rating:** 6
**Confidence:** 4

**Strengths:**

- The paper extends the scope of mono-lingual benchmarking. The LEPISZCZE benchmark can be used to track progress in developing mono-lingual models for Polish and cross-lingual models overall.
- All datasets but proprietary ones are distributed via HuggingFace hub for better accessability.
- The baseline movel evaluation is well documented via MLOps tools.

**Weaknesses:**

- The paper aggregates 13 existing datasets into a single benchmark. No novel datasets are introduced. The focus of chosen datasets is not well explained — it is clear, that the authors tried to account for a large number of datasets, but it is not clear how these particular datasets were chosen. These datasets do not cover a standart NLU / NLP scope or any domain-specific problem.
- Mono-lingual datasets sich as GLUE or SuperGLUE include a human evaluation as a reference. The LEPISZCZE benchmark lacks of human evaluation. The papper does not provide inisghts, whether a human baseline will be introduced as a starting points.
- The choice of models seems unclear. The only cross-lingual model used is a sentence model. All mono-lingual models are encoders. Why not use a cross-lingual encoder such as RemBERT, mBERT, XLM-R?
- There is no overall metric in the benchmark. To this end, it is not clear, how to choose best model.

**Additional Feedback:**

Please see the comments above.

**Clarity:**

- How the Fig.1 was designed, how is the choice of languages motivated? How the number of native speakers was estimated?
- What are the strategies to cope with standart splits problem and how are they implemented in the LEPISZCZE benchmark?
- How constiniuos benchmarking is implenented in the LEPISZCZE benchmark?

**Correctness:**

Some of the choices in the benchmark design are questionable.(i) the choice of datasets and (ii) the choice of models. Human evaluation is not a must, but could provide better understanding of model’s skills.  There is not aggregated metric in the benchmark, so it is not clear how exactly the models should be compared.

**Documentation:**

The documentations is sufficient. Model training is logged with MLOps tools, changes in datat are tracked with Version control tools.  However the benchmark includes a dataset under restricted access, which poses a significant limitation to future users.

**Ethics:**

No ethical issues.

**Relation To Prior Work:**

The paper is grounded on previous reserach in NLP benchmarking.

**Summary And Contributions:**

This papers offers a novel benchmark for Polish. The benchmark consists of 13 existing dataset. Experimental evaluation covers 5 mono- and cross-lingual models. The papers makes a solid contirbution to the NLP benchmarking area.

---

> ### Author Response · Authors · 2022-08-25
> **Authors' response to Reviewer 3SWT**
>
> **Q1**: The paper aggregates 13 existing datasets into a single benchmark. No novel datasets are introduced. The focus of chosen datasets is not well explained — it is clear, that the authors tried to account for a large number of datasets, but it is not clear how these particular datasets were chosen. These datasets do not cover a standart NLU / NLP scope or any domain-specific problem.
>
> **A1:** We want to emphasize that we introduced two new datasets: PAC (Polish Abusive Clauses) and Dialogue Acts (Diabiz.Kom). Moreover, almost all of chosen by us datasets (i.e., KPWr-NER, AspectEmo, PolEmo, DiaBiz.Kom, PAC, Political Advertising, and PSC) have been created by researchers in the CLARIN-PL-Biz project. Hence the annotation processes and inter-annotator agreements are properly described in the papers presenting them. We planned to gather many diverse tasks for Polish that cover different domains and tasks. However, since Polish is a low-resource language, our choice was limited. We ended up with text classification, natural language inference, and sequence tagging tasks.
>
> ---
>
> **Q2**: Mono-lingual datasets sich as GLUE or SuperGLUE include a human evaluation as a reference. The LEPISZCZE benchmark lacks of human evaluation. The papper does not provide inisghts, whether a human baseline will be introduced as a starting points. Human evaluation is not a must, but could provide better understanding of model’s skills.
>
> **A2:** Unfortunately, we did not have annotation resources to add to all datasets and information, and they were not added to each dataset paper as well.
>
> ---
>
> **Q3**: The choice of models seems unclear. The only cross-lingual model used is a sentence model. All mono-lingual models are encoders. Why not use a cross-lingual encoder such as RemBERT, mBERT, XLM-R?
>
> **A3:**  We are aware that a broader range of language models exists compared to the ones we included. We focused on providing solid baselines in the first iteration of the benchmark. Furthermore, we gave the users option to create a submission to the benchmark, according to instructions provided on the leaderboard page [https://lepiszcze.ml/submission/](https://lepiszcze.ml/submission/). Moreover, we encountered hardware limitation issues due to GPU memory limit and time-intensive computation.
>
> ---
>
> **Q4**: Some of the choices in the benchmark design are questionable.(i) the choice of datasets and (ii) the choice of models
>
> **A4:** The selection criteria for task and datasets are provided in the 3.1 section of updated the paper.
>
> ---
>
> **Q5**: There is no overall metric in the benchmark. To this end, it is not clear, how to choose best model. There is not aggregated metric in the benchmark, so it is not clear how exactly the models should be compared
>
> **A5**: Our goal is not to create a static leaderboard but rather a place to gather Polish NLP models for many tasks and allow users to create different subsets of tasks/datasets/models views. However, we added the averaged metrics to choose the best-performing model.
>
> ---
>
> **Q6**: How the Fig.1 was designed, how is the choice of languages motivated? How the number of native speakers was estimated?
>
> **A6**: We provided the missing citation for language statistics. We chose only language with at least one benchmark.
>
> ---
>
> **Q6**: What are the strategies to cope with standard splits problem and how are they implemented in the LEPISZCZE benchmark?
>
> **A6**: We described it in a new limitation section 5.
>
> ---
>
> **Q7**: How constiniuos benchmarking is implenented in the LEPISZCZE benchmark?
>
> **A7**:  We published leaderboard: [lepiszcze.ml](http://lepiszcze.ml/), which automatically parses submission into the results table. We additionally provided a Submission section ([https://lepiszcze.ml/submission/](https://lepiszcze.ml/submission/)) on our leaderboard page describing how one can add new scores to the benchmark. In addition, we log information about the dataset version during model evaluation, enabling us to create a different leaderboard for different dataset versions. Then different versions of the dataset could be treated as different datasets, and a separate leaderboard view can be generated.

---

> > ### Comment · Reviewer_3SWT · 2022-08-31
> > **2nd round: this reviewer keeps the score $6$**
> >
> > This reviewers appreciates the author's efforts in improving the paper. The reviewer keeps the original score $6$.

---

### Official Review · Reviewer_Zkmn · 2022-07-27
**Promising NLP Benchmark for Polish**

**Rating:** 7
**Confidence:** 3
**Correctness:** The main claims of the paper are corr…
**Clarity:** The paper is clear and concise.

**Strengths:**

* The contribution addresses an urgent and important need in NLP
* The benchmark is clearly described and well-documented
* The paper includes initial results on five different language models

**Weaknesses:**

While the core contribution of the paper is strong, I noticed a few ways the writing and organization could be improved before publication. My main suggestion is that the authors include more justification and reasoning for their choices. Currently, the paper clearly describes the choices made but not always *why* they were made.

I also have the following technical suggestions:
* In-text citations should be changed so they are parenthetical (this is probably just a matter of using a different latex command). For example, on page 2, `protocol Pineau et al. (2021)` should be `protocol (Pineau et al., 2021)`. This is a recurring error throughout the paper.
* The caption for Figure one says `NLU benchmarks`
* On page 3, I suggest a table for the benchmarks listed (rather than the long paragraph of authors)
* Is there a missing citation on line 91? (The text says `edition of Ethnologue?:`.)
* The header for 3.1.1 and 3.2 seem very similar and imply redundant sections
* On line 190, the sentence `We conclude agreements over the Internet daily` is unclear. Would "make agreements" or "accept agreements" fit?
* I suggest the authors include a discussion of limitations (or potential limitations) in section 5, as well as a discussion of ethical considerations.

**Additional Feedback:**

No additional feedback.

**Documentation:**

The datasets are well-documented, with links to all the different components.

**Ethics:**

The paper could be improved with a discussion of ethical considerations, perhaps in section 5.

**Relation To Prior Work:**

Section 2 describes related work in sufficient detail.

**Summary And Contributions:**

This paper presents an expansive NLP benchmark task for Polish NLP, modeled after GLUE (LEPISZCZE is the Polish word for glew). The benchmark addresses an urgent and important need to improve NLP for popular languages spoken around the world.

---

> ### Author Response · Authors · 2022-08-25
> **Authors' response to Reviewer Zkmn**
>
> **Q1**: My main suggestion is that the authors include more justification and reasoning for their choices. Currently, the paper clearly describes the choices made but not always *why t*hey were made.
>
> **A1:** We addressed that in section 3.1 Benchmark designing and construction process, in the updated version of the paper. We justified the choice of the datasets, models, and metrics. Moreover, we added the limitations of the KLEJ benchmark in the Related Work section.
>
> ---
>
> **Q2:**  Technical Suggestions
>
> - In-text citations should be changed so they are parenthetical (this is probably just a matter of using a different latex command). For example, on page 2, `protocol Pineau et al. (2021)` should be `protocol (Pineau et al., 2021)`. This is a recurring error throughout the paper.
>
>     **A:** We resolved issues with citations in the paper.
>
> - The caption for Figure one says `NLU benchmarks`
>
>     **A:** The figure caption is fixed.
>
> - On page 3, I suggest a table for the benchmarks listed (rather than the long paragraph of authors)
>
>     **A:** We added a table with benchmarks and references (appendix).
>
> - Is there a missing citation on line 91? (The text says `edition of Ethnologue?:`.)
>
>     **A:** We added the missing citation.
>
> - The header for 3.1.1 and 3.2 seem very similar and imply redundant sections
>
>     **A**: We rewrote section 3.
>
> - On line 190, the sentence `We conclude agreements over the Internet daily` is unclear. Would "make agreements" or "accept agreements" fit?
>
>     **A**: We updated the PAC dataset description.
>
> - I suggest the authors include a discussion of limitations (or potential limitations) in section 5, as well as a discussion of ethical considerations
>
>     **A:** We added limitations section 5 and moved ethical considerations from the Paper’s Checklist to the paper.
>
>
> ---
>
> **Q3**: The paper could be improved with a discussion of ethical considerations, perhaps in section 5.
>
> **A3:** It has been addressed in Paper’s Checklists. We moved the PII-related discussion about the PAC dataset to the paper.

---

> > ### Comment · Reviewer_Zkmn · 2022-09-02
> > **Response to authors**
> >
> > Thanks to the authors for responding to concerns from other reviewers as well as my own. In my view the concerns have been addressed and I still recommend the paper is accepted.

---

### Official Review · Reviewer_ksVA · 2022-07-28
**This is the way - lessons learned from designing and compiling LEPISZCZE, a comprehensive NLP benchmark for Polish**

**Rating:** 5
**Confidence:** 4

**Strengths:**

1. The paper fits the track well and addresses benchmarking for non-English languages, making a great attempt to create more resources for language model evaluation in Polish. The proposed benchmark also prompts the study of cross-lingual models' knowledge transfer abilities.
2. The benchmark's infrastructure, including data and experiment versioning, is an important consideration for evaluating models and updating the benchmark with new tasks or potential dataset edits.


**Weaknesses:**

1. The motivation behind choosing the benchmark datasets is unclear. In particular:
* Why the five KLEJ tasks are included and the others are not?
* I also wonder why the authors include the tasks (e.g., CDSC-E, NKJP POS) that do not seem to be that challenging for the considered models.

2. The authors state that they allow sorting submissions by various metrics (lines 158--159). However, it is unclear if the metrics are appropriate for all considered tasks and how the variability in the evaluation would reflect the progress on the benchmark. Besides, as far as I understand, some metrics are task-specific (e.g., https://huggingface.co/datasets/clarin-pl/2021-punctuation-restoration). Does the evaluation design account for them in the model ranking?

3. The authors provide recommendations on creating benchmarks for resource-lean languages in Section 3. Recommendations based on 3.1.2 and 3.1.3 do not have an empirical foundation and can be formulated as future work. At the same time, it is unclear how the benchmark design, apart from the infrastructure choices, can be replicated in other languages (e.g., lines 49--51).

4. The limitations of this work are paid little attention to. Could you please add the Limitations section?

**Additional Feedback:**

**Appendix**

The authors do not provide appendices and supplementary materials that should include the discussion on the related work (lines 88--89, 97--98).

**Suggestions**

Overall, in the revision, I would suggest:
1. providing the dataset examples, more details on the datasets, and the supplementary materials
2. clarifying the motivation behind the dataset choices and how the KLEJ limitations are addressed in the proposed benchmark
3. discussing the limitations and ethical considerations
4. continuing a discussion on the other mentioned comments, particularly the metric choice and model ranking.


**Questions**
Are there any plans on establishing the human baseline?

**Clarity:**

To me, the motivation of the work can be more focused on in the Introduction section. Contributions (1 and 3) and (2 and 4) can be combined. The authors could also provide more details on the dataset collection. To do this, the authors could remove content that is duplicated in the dataset cards (e.g., https://huggingface.co/datasets/laugustyniak/abusive-clauses-pl).

Based on the dataset description, the task formulation is not always precise. I would suggest adding examples for each dataset for better understanding.

**Correctness:**

As mentioned in the **Weaknesses** section, unifying task-specific metrics for all datasets can be confusing and lead to spurious conclusions about the model evaluation. Given that homogeneous metrics are used under such an approach, it is quite natural to rank the models by averaging their performance across all tasks. The model ranking on the potential LEPISZCZE leaderboard is not discussed, and how the ranking would change if the user wants to evaluate the models by another supported metric. The authors could also draw a few conclusions on the model performance across the domains.


The experimental setup lacks simple baselines, such as random guessing and simple linear classifiers, where applicable. I would also suggest adding more details on how the encoders are fine-tuned for the punctuation restoration task. However, the unified model finetuning and evaluation infrastructure is definitely a strong point.

**Documentation:**

**Reproducibility**

The authors provide sufficient detail to support reproducibility.

**Dataset documentation**
1. The datasets are not consistently described. For example, the authors provide results of the inter-annotator agreement for only a few datasets where human annotation was involved.
2. The authors do not characterize the target label distribution in the datasets. Are the train/val/test sets balanced?
3. AspectEmo, a 7-way classification dataset, includes a small number of test examples (292). Is the test set representative?
4. The license for some datasets, e.g., punctuation restoration, is not provided.
5. The dataset cards on the HuggingFace platform are not consistently filled.

**Ethics:**

The paper does not discuss the ethical considerations for the proposed datasets. I would suggest stating whether the work adheres to the ethical and responsible research standards (e.g., whether the datasets contain personal information, etc.)

**Relation To Prior Work:**

The work discusses related benchmarks for non-English languages and highlights the limitations of the closely related KLEJ benchmark. I would suggest straightforwardly stating how the proposed benchmark overcomes the KLEJ limitations.

**Summary And Contributions:**

This paper presents LEPISZCZE, a multi-task benchmark for evaluating neural models' Polish language understanding abilities. The benchmark includes five datasets from the KLEJ benchmark (released two years ago) and eight datasets mostly re-used from existing publications. LEPISZCZE covers various tasks across multiple domains, including text classification, sequence tagging, and sequence-to-sequence tasks. The authors conduct a series of extrinsic experiments using the proposed benchmark and monolingual and cross-lingual language models for Polish. The results indicate that the models perform strongly on many of the LEPISZCZE's tasks. At the same time, the work provides a flexible infrastructure that supports adding and documenting new tasks and models in a unified manner and practical recommendations on benchmarking.

---

> ### Author Response · Authors · 2022-08-25
> **Authors' response to Reviewer ksVA (Part I)**
>
> **Q1**: Why the five KLEJ tasks are included and the others are not?
>
> **A1:** The criteria for the task, dataset, and model selection are provided in Section 3.1 of the updated version of the paper.
>
> - CBD - In the near future, we plan to add this dataset to our benchmark. For now, we excluded this dataset due to limited computational resources - we could not provide a complete experimental analysis with an extensive hyperparameter search.
> - NKJP-NER is a simplified version of the typical NER task as it is defined as document classification in comparison to the standard sequence labeling task. Moreover, the dataset was prepared on the basis of a simplified typology of named entities. We provided the KPWR-NER dataset, which is appropriately defined as a sequence labeling task with fine-grained NE categories.
>
> We considered a few datasets for the regression task. However, in our opinion, text regression datasets in KLEJ are unsuitable for benchmarking.
>
> - According to the original paper that presented CDSC-R, the annotation was done by averaging rates from human annotators that assessed relatedness on a Likert scale (0-5). Such an assumption is very subjective and hence the Fleiss kappa score was low at 0.337.
> - Allegro Reviews consists of reviews collected from an e-commerce platform. Manual analysis of the samples showed that a lot of text reviews do not affect the scores (1-5) clearly.
>
> We hope to extend the benchmark by regression task in the future.
>
> ---
>
> **Q2:** I also wonder why the authors include the tasks (e.g., CDSC-E, NKJP POS) that do not seem to be that challenging for the considered models.
>
> **A2**: Regarding benchmark saturation, NKJP POS is the only PoS tagging dataset on the leaderboard, and we believe it is a simple but significant task for language models. On the other hand, the CDSC-E dataset provides a new domain of image captions.
>
> ---
>
> **Q3:** The work discusses related benchmarks for non-English languages and highlights the limitations of the closely related KLEJ benchmark. I would suggest straightforwardly stating how the proposed benchmark overcomes the KLEJ limitations.
>
> **A3:** In the updated version of the paper:
>
> - We gathered the limitations of KLEJ benchmark in our “Related Work” section
> - In section 3.1 "Benchmark designing and construction process," we described how our benchmark overcomes these limitations.
>
> ---
>
> **Q4:** *The authors state that they allow sorting submissions by various metrics (lines 158--159). However, it is unclear if the metrics are appropriate for all considered tasks and how the variability in the evaluation would reflect the progress on the benchmark. Besides, as far as I understand, some metrics are task-specific (e.g., [https://huggingface.co/datasets/clarin-pl/2021-punctuation-restoration](https://huggingface.co/datasets/clarin-pl/2021-punctuation-restoration)). Does the evaluation design account for them in the model ranking?*
>
> **Q5:** As mentioned in the Weaknesses section, unifying task-specific metrics for all datasets can be confusing and lead to spurious conclusions about the model evaluation. Given that homogeneous metrics are used under such an approach, it is quite natural to rank the models by averaging their performance across all tasks. The model ranking on the potential LEPISZCZE leaderboard is not discussed, and how the ranking would change if the user wants to evaluate the models by another supported metric. The authors could also draw a few conclusions on the model performance across the domains.
>
> **A4, A5**:  It is worth noting that our benchmark contains more than a single leaderboard table. We provided separate sub-leaderboards for different subsets of tasks/datasets. Hence, we can also rate the models based on different evaluation metrics, and we also give the user an option to compare models by choosing various metrics logged in the leaderboard. The models are evaluated mainly within classification setting using only the metrics designed for classification tasks. Currently, the benchmark generates a final ranking of language models by taking into account various metrics logged across different classification tasks (main leaderboard table). The final score of a model is computed only with respect to specific evaluation metric, as it might be inappropriate to mix different metrics altogether (homogeneity requirement). Nonetheless, it is possible to order and compare different language models on the main leaderboard by selecting specific evaluation metric as desired by the user (F1-macro / F1-micro / Accuracy / Precision and so on).

---

> > ### Author Response · Authors · 2022-08-25
> > **Authors' response to Reviewer ksVA (Part II)**
> >
> > **Q6:** The authors provide recommendations on creating benchmarks for resource-lean languages in Section 3. Recommendations based on 3.1.2 and 3.1.3 do not have an empirical foundation and can be formulated as future work. At the same time, it is unclear how the benchmark design, apart from the infrastructure choices, can be replicated in other languages (e.g., lines 49--51).
> >
> > **A6**: We added a section explaining the benchmark designing and construction process and how to replicate it in other languages in the updated version of the paper. Also, the recommendations are now framed as future work.
> >
> > ---
> >
> > **Q7**: The limitations of this work are paid little attention to. Could you please add the Limitations section?
> >
> > **A7:** We added a limitation section to this work (section 5).
> >
> > ---
> >
> > **Q8**: The experimental setup lacks simple baselines, such as random guessing and simple linear classifiers, where applicable. I would also suggest adding more details on how the encoders are fine-tuned for the punctuation restoration task.
> >
> > **A8:** We plan to add majority class baselines and simple classifiers with other text representations, such as BoW or static embeddings, to the leaderboard for comparison.
> > We formulate a punctuation restoration task as a sequence labeling task. The main goal is to predict whether, after a given token in a sequence, there exists a punctuation mark. The fine-tuning process is typical for sequence labeling tasks and very similar to POS tagging. For performance evaluation, we use a custom wrapper of the seqeval library, which calculates metrics for unit tags.
> >
> > ---
> >
> > **Q9**: Are there any plans on establishing the human baseline?
> >
> > **A9:** Unfortunately, we did not have any possibility to provide human baselines, and they were not added to each dataset paper.
> >
> > ---
> >
> > **Q10**: To me, the motivation of the work can be more focused on in the Introduction section. Contributions (1 and 3) and (2 and 4) can be combined.
> >
> > **A10:** We highlighted our motivation in the Introduction section and in a new section, “Benchmark designing and construction process” (3.1), to ensure the main motivation is clear for the readers. Moreover, we revisited the section “Contributions” and we combined all mentioned points into two major achievements, as suggested by the reviewer. Unfortunately, we did not have any possibility to provide human baselines, and they were not added to dataset descriptions in the paper.
> >
> > ---
> >
> > **Q11:** The authors could also provide more details on the dataset collection. To do this, the authors could remove content that is duplicated in the dataset cards (e.g., [https://huggingface.co/datasets/laugustyniak/abusive-clauses-pl](https://huggingface.co/datasets/laugustyniak/abusive-clauses-pl)).
> >
> > **Q12:** Based on the dataset description, the task formulation is not always precise. I would suggest adding examples for each dataset for better understanding.
> >
> > **Q13:** The datasets are not consistently described. For example, the authors provide results of the inter-annotator agreement for only a few datasets where human annotation was involved.
> >
> > **Q14**: The authors do not characterize the target label distribution in the datasets. Are the train/val/test sets balanced?
> >
> > **Q15**: The license for some datasets, e.g., punctuation restoration, is not provided.
> >
> > **Q16:** The dataset cards on the HuggingFace platform are not consistently filled.
> >
> > **A11-A16:** We unified the dataset descriptions by including the key information: short dataset description, task description, domains, measurements, data sample, data splits, class distribution (for each subset: train/val/test), BibTeX citation, licenses, and links to Hugging Face hub, sources, and original papers, if available. We decided to publish it on the Hugging Face hub and on the LEPISZCZE website [https://lepiszcze.ml/datasets/](https://lepiszcze.ml/datasets/). We also contributed to the KLEJ datasets by adding dataset cards. We decided not to put the dataset descriptions in our paper due to their excessive size. Moreover, users can use the Dataset Preview tool on Hub to view the examples easily or utilize our tool called Dataset Explorer (we will include all the datasets as soon as possible) linked on the benchmark website [https://huggingface.co/spaces/clarin-pl/datasets-explorer](https://huggingface.co/spaces/clarin-pl/datasets-explorer). We added the missing license of punctuation restoration dataset.

---

> > > ### Author Response · Authors · 2022-08-25
> > > **Authors' response to Reviewer ksVA (Part III)**
> > >
> > > **Q17:** AspectEmo, a 7-way classification dataset, includes a small number of test examples (292). Is the test set representative
> > >
> > > **A17:** We are aware that some test sets may not be fully representative. However, it is a common issue for low-resource languages such as Polish language, where access to annotated data is very limited. Nevertheless, the Authors of the AspectEmo dataset (who are also members of the CLARIN-PL consortium) are working on an improved version of that dataset. We will update all experiments as soon as an updated version of this dataset is available.
> > >
> > > ---
> > >
> > > **Q18:** The paper does not discuss the ethical considerations for the proposed datasets. I would suggest stating whether the work adheres to the ethical and responsible research standards (e.g., whether the datasets contain personal information, etc.)
> > >
> > > **A18:** It has been addressed in Paper’s Checklists, we moved the PII-related discussion about the PAC dataset to the paper.
> > >
> > > ---
> > >
> > > **Q19:** The authors do not provide appendices and supplementary materials that should include the discussion on the related work (lines 88--89, 97--98).
> > >
> > > **A19:** We added appendix sections as our supplementary material. The leaderboard is now accessible via the website [https://lepiszcze.ml/](https://lepiszcze.ml/datasets/)

---

### Author Response · Authors · 2022-08-25
**Authors' summary response to all reviewers with outlined changes**

We want to thank all the reviewers for their detailed feedback. We carefully followed all the raised corners. We hope that these changes address your major concerns. In case of any doubts, we ask for your feedback. Here, we summarize our major improvements in comparison to the first version:

- We published the leaderboard webpage. [https://lepiszcze.ml](https://lepiszcze.ml/), where we gathered various metrics, rankings, sub-leaderboards, and dataset cards, and we added guidelines for adding a submission to the benchmark.
- We unified dataset cards by covering all of the essential information. We extended the cards by class distributions. We also added missing ones. Moreover, we made a contribution to KLEJ datasets on the HuggingFace hub.
- We added dataset explorer on the Hugging Face spaces webpage, which consists of more possibilities to explore benchmark datasets than the default Hugging Face preview. [https://huggingface.co/spaces/clarin-pl/datasets-explorer](https://huggingface.co/spaces/clarin-pl/datasets-explorer)
- We have updated the paper accordingly. We highlighted differences between KLEJ and our Benchmark and justified our choices for benchmark creation (datasets/model/task selection). We have rewritten section 3.1, which is more clarified now; we formulated it so that it is more straightforward to reproduce the benchmark in other languages. We added the limitation of the previous polish NLP benchmark, KLEJ, in the related work section. We have also added a Limitation section to discuss the current shortcomings of our benchmark. Finally, we fixed some of the dataset descriptions to add some clarity.
- We added Appendix in the form of supplementary material.
    - We added information about the reviewed benchmarks in the Introduction section.
    - We included information about the hyperparameters search space for the hyperparameter search procedure.
    - We added missing results tables consisting of the different metrics we analyzed.

---

### Meta-Review · Area_Chair_idRj · 2022-09-09

**Recommendation:** Accept
**Confidence:** 5

**Metareview:**

This work is about a multi-task benchmark for evaluating Polish NLU, called LEPISZCZE. This curates 5 datasets from an existing benchmark and adds eight more with a bit of reusage. There are a variety of domains and tasks covered. The authors conduct various experiments on the benchmark to evaluate performance of monolingual and crosslingual language models. Models perform pretty well on this benchmark already. Overall, the authors should include more motivation on exactly how these benchmarks were chosen and no new datasets are contributed. However, the benchmark clearly has value and creating benchmarks for more languages is important work for the community. Further, authors & reviewers had excellent and thorough discussion on reviewer concerns.

---

### Decision · Program_Chairs · 2022-09-16

Accept